# GluR2$^Q$ and GluR2$^R$ AMPA Subunits are not Targets of lypd2 Interaction

**Anna Lauriello[1¤a], Quinn McVeigh[2¤b¤c], Rou-Jia Sung[2]***

**1** Department of History, Carleton College, Northfield, MN, United States of America, **2** Department of Biology, Carleton College, Northfield, MN, United States of America

¤a Current address: Sidney Kimmel Medical College, Thomas Jefferson University, Philadelphia, PA, United States of America
¤b Current address: Cancer Program, Broad Institute, Cambridge, Massachusetts, United States of America
¤c Current address: Department of Medical Oncology, Dana Farber Cancer Institute, Boston, Massachusetts, United States of America
* rsung@carleton.edu

**Data Availability Statement:** All relevant data are within the paper and its Supporting Information files.

**Funding:** The author(s) received no specific funding for this work.

## Abstract

A large family of prototoxin-like molecules endogenous to mammals, Ly6 proteins have been implicated in the regulation of cell signaling processes across multiple species. Previous work has shown that certain members of the Ly6 family are expressed in the brain and target nicotinic acetylcholine receptor and potassium channel function. Structural similarities between Ly6 proteins and alpha-neurotoxins suggest the possibility of additional ionotropic receptor targets. Here, we investigated the possibility of lypd2 as a novel regulator of AMPA receptor (AMPAR) function. In particular, we focused on potential interactions with the Q/R isoforms of the GluR2 subunit, which have profound impacts on AMPAR permeability to calcium during neuronal stimulation. We find that although lypd2 and GluR2 share overlapping expression patterns in the mouse hippocampus, there was no interaction between lypd2 and either GluR2$^Q$ or GluR2$^R$ isoform. These results underscore the importance of continuing to investigate novel targets for Ly6 interaction and regulation.

## Introduction

Regulation of cellular signaling processes often begins at the plasma membrane, in which regulatory proteins interact with transmembrane receptors to modulate their response following an extracellular signal. Dysregulation of these processes has significant negative consequences for cellular and organismal function, with aberrant signaling processes implicated in dozens of human disease states [1], underscoring the need for the identification and characterization of novel regulators of cell signaling. The Ly6 protein family has been shown to be a prolific source of potential regulators for cell signaling, with recent work in the last few decades implicating Ly6 proteins in the modulation of cell-cell communication across multiple systems [2–13]. However, the number of characterized Ly6 proteins remains low, despite the large number of candidates in the family (40+ genes in mammalian systems) [14]. Averaging 15-20kDa in size, Ly6 proteins are generally found within the extracellular space, either as a secreted protein or

**Competing interests:** The authors have declared that no competing interests exist.

(in the majority of cases) by being physically tethered to the outer leaflet of the plasma membrane by a post-translational C-terminal GPI anchor modification [6, 14–16]. The Ly6 proteins adopt a three-finger fold, comprised of three variable sequence loops stabilized by a "crown" of 5–6 highly conserved disulfide bonds [13, 17]. This three-finger fold is also found in α-neurotoxins commonly found in elapid snake venoms, such as α-bungarotoxin [16, 18]. These α-neurotoxins primarily exert their effects by binding to orthosteric sites on their target receptor and inhibiting the conformational changes necessary for receptor function. Given their structural similarities, early explorations of Ly6 function hypothesized that Ly6 proteins may also target ionotropic receptors in a similar fashion as α-neurotoxins [19, 20]. Indeed, the first studies done on lynx1 and lynx2 demonstrated that these Ly6 proteins also target nicotinic acetylcholine receptors (nAChRs)—α-bungarotoxin targets the α7 nAChR while lynx1 and lynx2 target both α4β2 and α7 nAChRs—by altering the kinetics of receptor desensitization and receptor stochiometry, with implications for learning and memory [19–22]. Recent work has shown that Ly6h also modulates nicotinic acetylcholine receptor function via a different mechanism—by lowering levels of receptor found at the plasma membrane and thus altering the maximal response of these receptors to agonist [23–25]. Outside of nicotinic acetylcholine receptors, these α-neurotoxins can also interact with other ionotropic receptors, such as L-type calcium channels and voltage-gated potassium channels [26]. Could Ly6 proteins also modulate the function of such a diverse array of receptors? Current work has identified only a few non-nicotinic acetylcholine receptor targets for Ly6 proteins, including the Shaker-type potassium channels [27], CD3ζ chains of the T cell receptor signaling complex (TCR) [28], TBK1 [29], and EGFR/PDGFR heterodimer [30]. This work demonstrates that Ly6 proteins can modulate the function of non-nicotinic acetylcholine receptors, opening up possibilities of identifying other novel targets of Ly6-mediated regulation. Moreover, the role of Ly6 proteins in regulating neuronal excitability is conserved across multiple organisms (sss (*sleepless*) regulates sleep in Drosophila [27, 31] and odr-2 mediates odor sensing in *C. elegans* [32, 33]), suggesting other Ly6 proteins may also be involved in similar processes.

In mammalian systems, the majority of excitatory neurotransmission is mediated by ionotropic glutamate receptors. Specifically, AMPA receptors (AMPARs) are highly expressed in the brain, particularly in the hippocampal region [34]. AMPARs are synthesized, folded, and assembled in the endoplasmic reticulum, where they must undergo stringent quality control prior to export to the Golgi and ultimately to the cell surface [35–37]. Several auxiliary proteins (such as TARPs, including stargazin, and CNIHs) and small interacting regulatory proteins (such as GRIP2/ABP and PICK1) have already been identified as playing key roles in AMPAR trafficking to the cell surface [38–41]. In particular, TARPs and CNIHs are both large families of AMPAR auxiliary proteins that are localized to the plasma membrane; recent structures of AMPARs in complex with these transmembrane auxiliary proteins reveal that due to their small size and proximity to the plasma membrane, many of the functional interactions with the AMPAR are with the linker regions between the ligand binding domain (LBD) and transmembrane domain (TMD) or the LBD and the N-terminal domain (NBD) [39, 40, 42–48]. Given the importance of AMPARs in modulating excitatory neurotransmission in mammalian brains and the significant number of the mammalian Ly6 proteins that also play roles in regulating neuronal excitability and neuronal function, this led us to hypothesize: could Ly6 proteins also be regulators of AMPA receptor function?

As the first step for considering that question, we investigated the possibility of whether lypd2, a Ly6 protein predicted to be GPI-anchored and reported to be expressed in the hippocampus, could interact with GluR2 AMPA receptors using co-immunoprecipitation in HEK-293 cells [51]. AMPA receptors are tetrameric and can exist as heteromeric combinations of predominantly GluR1/GluR2, GluR2/GluR3, or GluR1/GluR2/GluR3/GluR2 or as homomers

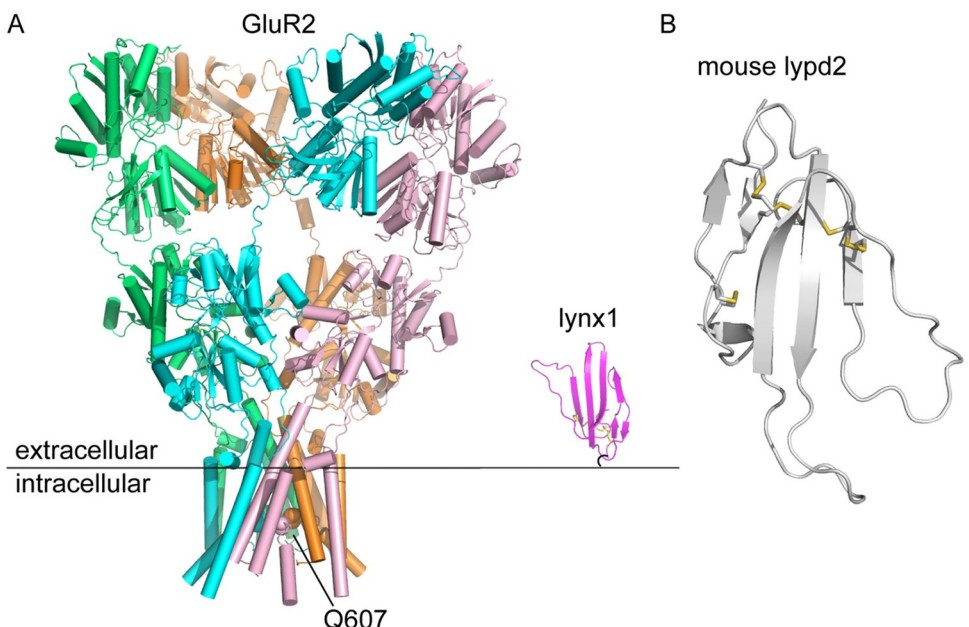

**Fig 1. Model for potential Ly6 interaction with GluR2.** A) Structures of GluR2 (3KG2) and lynx1 (2L03) shown as cartoon relative to outer leaflet of the plasma membrane [17, 49]. Each GluR2 subunit of the tetramer is shown in a different color; R607Q editing site on GluR2 is shown as spheres (in 3KG2 it has been edited to Q, as indicated), C-terminal GPI anchor on lynx1 is shown as a black curved line. B) Alphafold prediction of structure of mouse lypd2 (resides 21–100, omitting predicted N- and C-terminal signal sequences) [50]. Disulfide bonds are shown as gold sticks.

of either GluR1 or GluR2 [52]. Receptor composition has significant implications for receptor function and regulation, with each receptor combination showing differences in calcium permeability, receptor trafficking and assembly [53, 54]. Given this varied molecular landscape of AMPA receptor function and the lack of information regarding potential for regulation by Ly6 proteins, we chose to focus our study on homomeric GluR2$^R$ or Glu2R$^Q$ receptors, which differ in a single amino acid at position 607 located in the pore region of the receptor (Fig 1A). As a simpler system, homomeric assemblies allow us to directly assess interactions between just two proteins (the homomer and lypd2) and limits the number of possible interacting combinations that could occur with heteromeric systems. Moreover, both isoforms have been shown to have unique biochemical properties of their own. Receptors containing GluR2$^R$ remain calcium impermeable, while incorporation of GluR2$^Q$ in either heteromeric or homomeric forms retains calcium permeability; alternative splicing events during GluR2 maturation result in the Q607R conversion in the pore region to render 'mature' GluR2 calcium-impermeable [55–57]. In addition to effects on the mature receptor, previous work has also shown that the R607Q conversion has impacts on immature receptor folding and assembly, with changes in ER export kinetics for each isoform [57]. Moreover, both isoforms are differentially regulated by auxiliary proteins, including type II TARPs [53, 56–61]. In comparisons between GluR2$^R$ and GluR2$^Q$ isoforms that differ in sequence only at the 607 position, γ-5 lowered glutamate affinity and accelerated desensitization of only the GluR2$^R$ isoform, while γ-7 enhanced steady state currents in the presence of CTZ and glutamate for GluR2$^R$ and not GluR2$^Q$ [61]. As noted earlier, structures of γ-5 in complex with GluR2$^Q$ suggest the effects of this type II TARP on AMPAR function are mediated by interactions between its extracellular loops with the LBD and/or LBD-TMD linker [48]. Given this isoform-specific regulation of GluR2$^R$ vs GluR2$^Q$ by small regulatory proteins in close proximity to the linker/LBD regions of the receptor and the

biochemical simplicity of working with a homomeric system, we chose to focus our efforts on whether lypd2 could interact with homomeric GluR2$^R$ or GluR2$^Q$. As shown in Fig 1A, the relative sizes and positioning of the GluR2 tetramer relative to lynx1 (one of the few Ly6 proteins with an experimentally determined structure, shown here for accuracy in scale and structure; Fig 1B shows a predicted structure of lypd2 as modeled in AlphaFold, with significant structural similarities with lynx1) suggest that a Ly6 protein is within reach of the LBD and/or the linker regions between the LBD and the TMD (potentially the NTD as well, depending on conformational flexibility of the receptor) [50]. This physical range of functional impact is quite similar to the contact regions seen for structures of the GluR2-TARP γ-2, GluR2-TARP γ-5, and GluR2-CNIH3 complexes [42, 43, 48]. Given the internal positioning of Q607 (which is buried in the transmembrane domain and directly faces the inside of the pore), it is unlikely that any mechanism of regulatory protein interaction (either by auxiliary or Ly6 proteins) would be through direct contacts with that residue. The results of our experiments showed that lypd2 does not interact with homomeric GluR2$^R$ or GluR2$^Q$ AMPA receptors. Although we were unable to identify an interaction between these Ly6 proteins and AMPA receptors, we hope this work will prompt the field to consider investigating alternative targets of Ly6 regulation.

## Materials and methods

### DNA constructs

Rat cacng2 (stargazin; NM_053351.1) was synthesized and subcloned into pcDNA3.1+/C-(K)-DYK using CloneEZ strategy (Genscript). Rat lypd2 (NM_001130545.1) and mouse lypd2 (NM_026671.1) were synthesized and subcloned into pcDNA3.1(+)-myc-His A using HindIII/EcoRV sites (Genscript). The FLAG epitope is located immediately N-terminal to the predicted GPI attachment residue (as predicted using http://mendel.imp.ac.at/gpi/gpi_server. html [62]). GFP-GluR2$^R$ plasmid was a kind gift from R. Malinow. GFP-GluR2$^Q$ plasmid was generated from GFP-GluR2$^R$ plasmid using site-directed mutagenesis using the indicated primers: R607Q_f: `CCT TGG GTG CCT TTA TGC AAC AAG GAT GCG ATA TTT CGC`; R607Q_r: `GCG AAA TAT CGC ATC CTT GTT GCA TAA AGG CAC CCA AGG`.

### RT-PCR

Mouse whole brain and hippocampal cDNA were purchased from Zyagen (Mouse C57 Brain and Hippocampus cDNA). RT-PCR was carried out for GAPDH and lypd2 using the indicated primers: mGAPDH-QF: `5'-GTTGTCTCCTGCGACTTCA-3'`; mGAPDH-QR: `5'-GGTGGTCCAGGGTTTCTTA-3'`; mlypd2 QF: `5'-GCATCCAACTGTGTCACCAC-3'`; mlypd2 QR: `5'-GTCAGAATTGCAGCAGGACA-3'`. Anticipated sizes for GAPDH and lypd2 products were 184bp and 192bp, respectively.

### Cell culture

HEK-293T cells (ATCC, CRL-1573) were maintained at 37˚C and 5% CO2 in culture medium consisting of 10% fetal bovine serum (Omega), 1% penicillin/streptomycin (Corning), and 1% l-glutamine (Sigma) in low-glucose DMEM with 2mM l-glutamine (Corning). Cells were grown to 60–80% confluence for transfection with X-tremeGENE HP reagent (Roche) at a 2:1 ratio of transfection reagent to DNA in Opti-MEM (Thermofisher). Transfection mixture was removed 24h after transfection and replaced with normal growth medium. Cells were allowed to recover for 24h in normal grown medium prior to immunostaining or immunoprecipitation protocols below.

## Immunostaining

HEK-293T cells were plated on glass coverslips coated with poly-D-lysine (Sigma) and cultured for 24h before transfection. Transfection was done as described above. Cells were rinsed with ice-cold PBS and fixed in 4% formaldehyde/PBS for 10min at room temperature. Cells were blocked for 1h at room temperature in 10% normal goat serum (Thermofisher) and then sequentially incubated in primary antibodies (overnight at 4C with rabbit α-GFP (Thermofisher, A11122) or mouse α-FLAG (Thermofisher, MA191878)). Labeled cells were washed three times with ice-cold PBS and incubated in secondary antibody for 1h at room temperature and DAPI (Sigma) for 1min at room temperature. Cells were mounted onto glass slides in ProLong Glass Anti-fade mountant (Thermofisher). Fluorescently conjugated secondary antibodies used were Alexa Fluor goat anti-mouse 568 and Alexa Fluor goat anti-rabbit 488 (Thermofisher). Cells were imaged on a Zeiss AxioImager at 40X magnification. Figures show representative imaged from at least three independent experiments for each set of transfected conditions.

## Co-immunoprecipitation (Co-IP)

Proteins were extracted with "+SDS" lysis buffer (10mM Tris pH 7.5, 100mM NaCl, 5mM EDTA, 1% Triton X-100, 0.05% SDS, Complete protease inhibitor (Roche)). Protein concentrations were quantified via BCA Protein Assay (Thermofisher). 75 μg of total protein was reserved for input and 750μg for Co-IP. Co-IP samples were rocked for a minimum of 4 hours with 12.5 μl Protein G magnetic beads (NEB) at 4˚C. Samples were washed 3X with a Co-IP +SDS wash buffer (10mM Tris pH 7.5, 100mM NaCl, 5mM EDTA, 0.05% Triton X-100, 0.05% SDS, Complete protease inhibitor (Roche)), then stored at -20C in 1X SDS Sample Buffer. "-SDS" lysis and wash buffers were the same as above except they were made without SDS; "PBS" lysis and wash buffers were phosphate buffered saline (PBS) purchased from Corning.

## Western blotting

Samples were heated for 10 minutes at > 90˚C. Proteins were run on 4–20% gels (Bio-Rad Mini-PROTEAN® TGX Stain-Free™ Protein Gels) in 1X TGS buffer (Bio-Rad) and transferred using a semi-dry transfer (Bio-Rad Trans-Blot® Turbo Transfer System) to a 0.2μm nitrocellulose membranes (Bio-Rad). Blocking was done in a 5% skim milk in 1X TBST. Membranes were blotted using primary antibodies against the mouse α-FLAG antibody (Thermofisher, MA191878), rabbit α-actin (Thermofisher, MA5-32479), or rabbit α-GFP antibody (Thermofisher, A11122). Figures show representative immunoblots from at least three independent experiments for each set of transfected conditions.

## Results

AMPA receptors are highly expressed throughout the brain, particularly in the hippocampus. We reasoned that Ly6 proteins that are also highly expressed in the brain would have a higher likelihood of interacting with AMPA receptors. Using RT-PCR analysis, we confirmed expression of lypd2 in mouse whole brain and hippocampal cDNA samples (Fig 2A). HEK 293 cells have been extensively demonstrated to be suitable for the expression of functional AMPA receptors in a heterologous system. In order to assess if they were a suitable heterologous expression system for lypd2, we performed a series of expression tests using constructs generated from mouse and rat genomes. We found that mouse and rat lypd2 expressed well and showed punctate expression at the cell surface, which is in agreement with previously

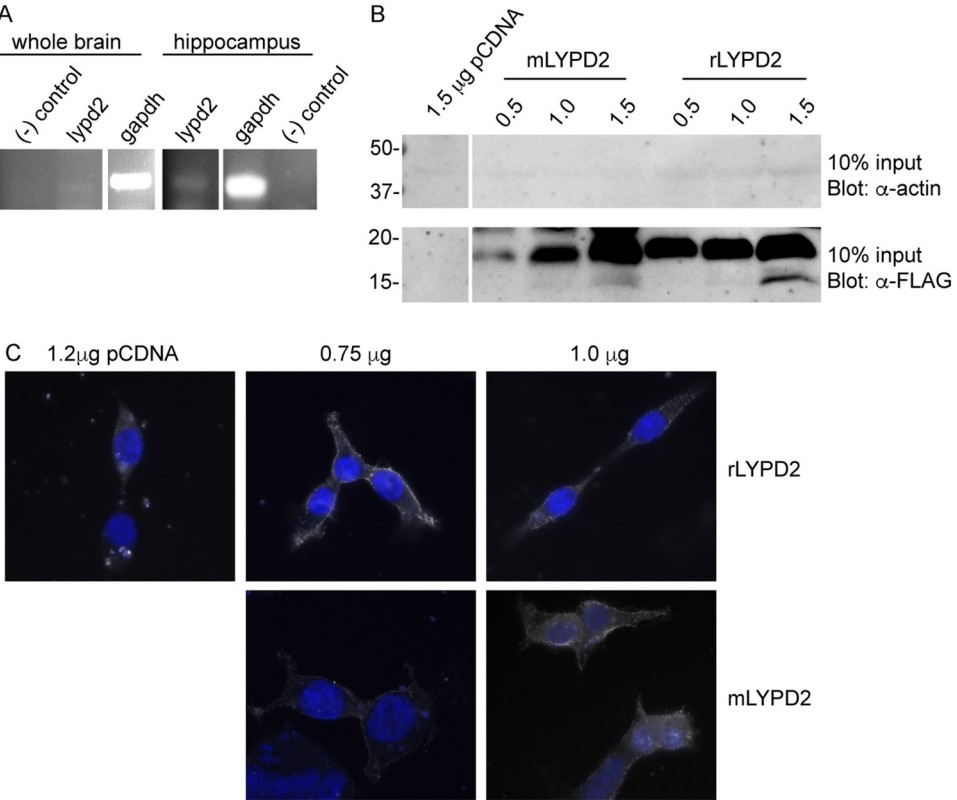

**Fig 2. Expression of rodent lypd2 in whole brain, hippocampus, and HEK-293T cells.** (A) RT-PCR showing expression of lypd2 in mouse whole brain and hippocampal cDNA (n = 3). (B) Expression of FLAG-tagged *M. musculus* lypd2 and *R. norvegicus* lypd2 in HEK-293T cells as measured by Western Blot (n = 3). Cells were transfected with indicated amounts of DNA for each construct and blotted for presence of C-terminal FLAG epitope. Empty vector (pCDNA) was used as a control. (C) Representative images of cell surface expression of lypd2 constructs visualized using immunofluorescence under non-permeabilizing conditions (n = 3). HEK-293T cells were transfected with the indicated amounts of each construct and stained for presence of C-terminal FLAG epitope. Blue: DAPI, white: FLAG.

established cellular localization patterns for other Ly6 proteins (Fig 2B and 2C) [23, 63]. Having established expression conditions for lypd2, all remaining experiments were conducted using the rat isoforms for these Ly6 proteins. As shown in Fig 3A, co-immunoprecipitation experiments with GluR2 homomeric AMPA receptors did not show any interaction with lypd2. These results were consistent for both GluR2$^R$ and GluR2$^Q$ isoforms.

In order to determine if our buffer conditions were appropriate for detecting protein-protein interactions with GluR2, we performed a pulldown using GluR2 and stargazin. Stargazin has been previously shown to interact with and regulate the activity of GluR2, and served as a positive control for our pulldown conditions [64, 65]. As shown in Fig 3B, we were able to successfully pulldown stargazin using GluR2$^Q$. Having established that our initial buffer conditions were suitable for pulling down GluR2-interactors, we considered the possibility that these buffer conditions were too stringent for detecting potentially weaker interactions between GluR2 and lypd2. We repeated our lypd2 GluR2$^Q$ pulldown with alternative buffers (-SDS from ours; PBS) under less stringent conditions [66]. As shown in Fig 3C, less stringent buffers continued to result in no pulldown, indicating that the lack of interaction seen is not due to our buffer conditions.

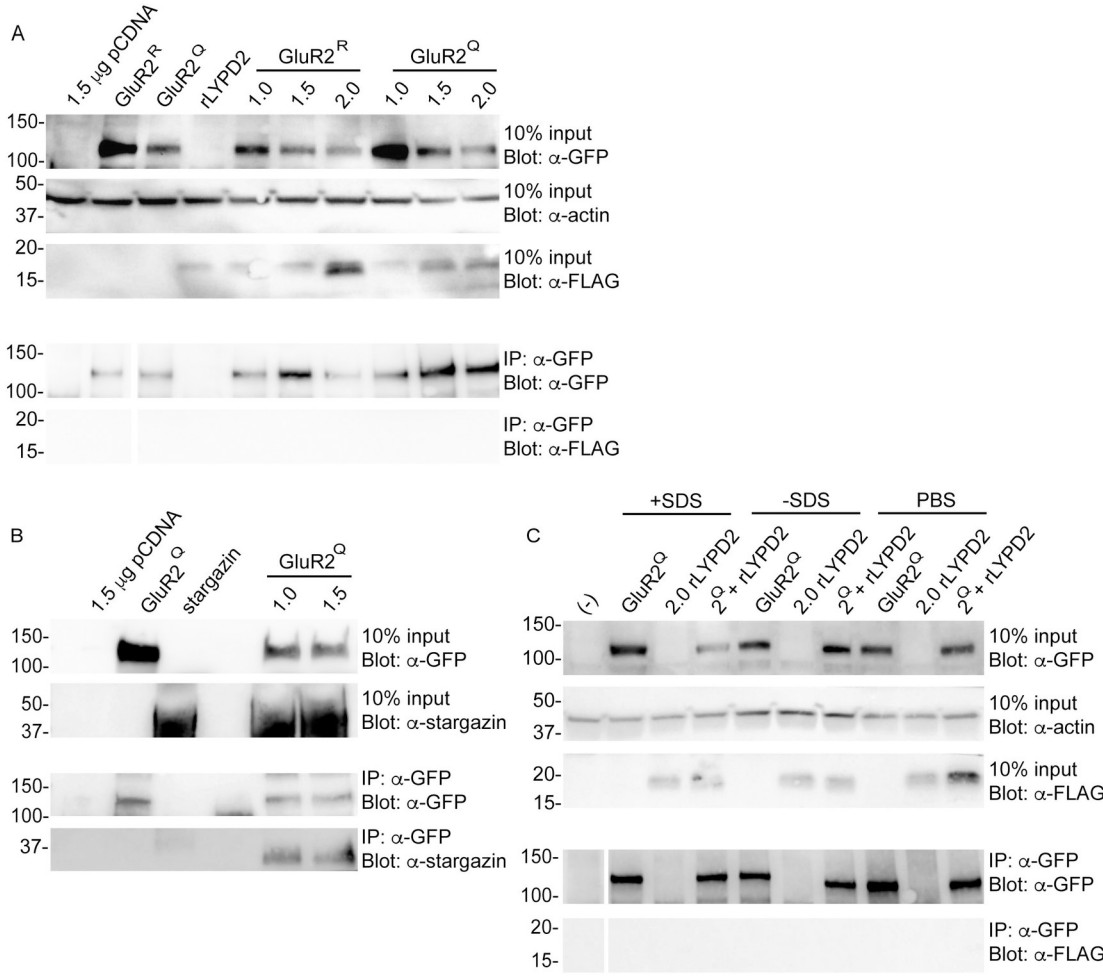

**Fig 3. Lypd2 does not interact with GluR2$^Q$ or GluR2$^R$.** (A) Representative Western Blots of lypd2-FLAG co-immunoprecipitated with GFP-GluR2$^Q$ or GFP-GluR2$^R$ (n = 4). (B) Representative Western Blots of stargazin co-immunoprecipitated with GFP-GluR2$^Q$ (n = 4). (C) Representative Western Blots of lypd2-FLAG co-immunoprecipitated with GFP-GluR2$^Q$ prepared with different lysis and wash buffers (see Materials and Methods; n = 3). Top panels show 10% input blotted with the indicated primary antibodies; bottom panels show pulldown samples blotted with the indicated primary antibodies. HEK-293T cells were transfected with the indicated μg of DNA for each construct; cells were transfected with 0.5 μg of receptor DNA and the indicated amounts of stargazin or lypd2-FLAG DNA.

## Discussion

The Ly6 protein family has been implicated in the regulation of cell-cell communication pathways across multiple model systems. In several examples, the molecular mechanism underlying this regulatory function is due to alterations in the trafficking and function of ionotropic receptors such as nicotinic acetylcholine receptors and potassium channels. Here, we explored an alternative approach towards identifying potential new ionotropic receptor targets of Ly6 regulation. We focused on AMPA receptors as a possible target due to their role in mediating neuronal communication, the well-established precedence for regulation of AMPA receptor function via small regulatory proteins, and their highly dense expression in particular regions of the brain (including the hippocampus) where lypd2 is also expressed [34, 35, 37, 38, 54].

As noted in the introduction, we focused our efforts on the R/Q isoforms of GluR2 due to previously established work indicating isoform-specific regulation by auxiliary and/or small regulatory proteins [61]. Although our experiments indicate that we were unable to isolate

stable complexes between either $GluR2^Q$ or $GluR2^R$ homomeric receptors with lypd2, they do not rule out the possibility of a Ly6-AMPAR interaction with other receptor combinations (GluR1 homomers, GluR1/GluR2, GluR2/GluR3, for example), with other GluR2 isoforms that vary in the intracellular or extracellular regions (flip/flop, R/G editing, differences in glycosylation patterns, etc), or in the presence of an additional binding partner in a trimeric complex [35, 52, 58, 67, 68]. In particular, previous work has demonstrated that native AMPARs in the brain are almost always associated with auxiliary subunits (such as γ-8 and CNIH2); therefore, if lypd2 were interacting with native AMPARs in the hippocampus it would likely be doing so in the context of a pre-existing complex of AMPAR with auxiliary subunits [52]. Future explorations of potential Ly6-AMPAR interactions could account for this complexity by including these additional auxiliary subunits in the transfection system. Moreover, additional work is also necessary to determine whether the observed overlaps in hippocampal expression for AMPARs and lypd2 (previous studies and this work, respectively) correlate with physical co-localization between lypd2 and AMPARs, a biological feature characteristic of interacting partners.

Although we were unsuccessful in identifying new targets for Ly6 regulation in this study, the importance of considering alternative strategies for studying the targets of this large and diverse protein family remains. Many of these Ly6 proteins and their associated molecular targets were identified as outcomes of phenotype-based screens; however, we posit that perhaps a biochemical approach may provide valuable information towards identifying the regulatory network of Ly6 interactors beyond the current few (such as using Ly6 proteins as bait to pull down novel prey interacting proteins that could be identified using mass spectrometry analysis) [69, 70]. The large size of this protein family suggests there could be additional Ly6 regulatory targets waiting to be identified.

## Supporting information

**S1 Raw images.**
(PDF)

## Acknowledgments

We thank R. Malinow for the kind gift of the $GFP-GluR2^R$ plasmid.

## Author Contributions

**Conceptualization:** Rou-Jia Sung.

**Data curation:** Rou-Jia Sung.

**Formal analysis:** Anna Lauriello, Quinn McVeigh, Rou-Jia Sung.

**Investigation:** Anna Lauriello, Quinn McVeigh, Rou-Jia Sung.

**Methodology:** Anna Lauriello, Quinn McVeigh, Rou-Jia Sung.

**Project administration:** Rou-Jia Sung.

**Supervision:** Rou-Jia Sung.

**Validation:** Rou-Jia Sung.

**Writing – original draft:** Rou-Jia Sung.

**Writing – review & editing:** Anna Lauriello, Quinn McVeigh.

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
