## [Decision Letter · Decision Letter 0]

12 Jul 2022

PONE-D-22-12127GluR2Q and GluR2R AMPA Subunits are not Targets of lypd2 InteractionPLOS ONE

Dear Dr. Sung,

Thank you for submitting your manuscript to PLOS ONE. After careful consideration, we feel that it has merit but does not fully meet PLOS ONE’s publication criteria as it currently stands. Therefore, we invite you to submit a revised version of the manuscript that addresses the points raised during the review process.In your revised manuscript please address as fully as possible the detailed and constructive comments and criticisms of Reviewer 2. 

We look forward to receiving your revised manuscript.

Kind regards,

Israel Silman

Academic Editor

PLOS ONE

Journal Requirements:

Reviewers' comments:

Reviewer's Responses to Questions

**Comments to the Author**

1. Is the manuscript technically sound, and do the data support the conclusions?

Reviewer #1: Yes

Reviewer #2: Partly

2. Has the statistical analysis been performed appropriately and rigorously? 

Reviewer #1: I Don't Know

Reviewer #2: No

3. Have the authors made all data underlying the findings in their manuscript fully available?

Reviewer #1: Yes

Reviewer #2: Yes

4. Is the manuscript presented in an intelligible fashion and written in standard English?

Reviewer #1: Yes

Reviewer #2: Yes

5. Review Comments to the Author

Reviewer #1: The task formulated by the authors - to detect novel receptors with which may interact the proteins of the Ly6/uPAR family - is well justified. They correctly started by emphasizing the well-known common three-finger folding of these proteins and of the three-finger alpha-neurotoxins from the snake venoms. However, they did clearly mention that there are two classes of the Ly6/uPAR proteins - secreted like SLURP-1 and membrane-bound by the GPI anchor like Lynx1. They gave the reasons why they have chosen lypd2 (because there are data on its colocalization with the AMPA receptors) but did not mention that lypd 2 is membrane-bound.

It is well-known that among the major targets of the Ly6/uPAR proteins are nicotinic acetylcholine receptors (nAChR) and the authors correctly give the references when as additional targets were found potassium channels and EGFR/PDGFR receptors.

The authors indeed demonstrated the colocalization in hippocampus of the lypd2 and of the calcium-conducting and non-conducting forms (Q/R isoforms of the GluR2 subunit) of the AMPA receptors, but found no interaction with their single chosen representative of the Ly6/uPAR family.

The authors made several mistakes in the introduction concerning snake venom neurotoxins:

line 49: major binding site of the snake venom alpha-neurotoxins is the orthosteric binding site (where agonists are bound), but not the allosteric sites (where mainly other allosteric positive or negative modulators are attached)

line 52: give references

line 53: alpha -bungarotoxin does not inhibit alpha4beta2 receptors , and instead of alpha7beta1 should be written " alpha7beta2".

line 20 and through the whole text authors use the term "ly6 proteins" , although most often in the literature are used Ly6 or Ly6/uPAR proteins.

------

Reviewer #2: The paper explores a possible interaction between a ly6 protein lypd2 and AMPA receptors using co-immunoprecipitation assays. It is an interesting topic certainly worth investigating. The result is negative which is in itself worth noting. The experiments are very basic, but the bigger problem is the experimental design. Overall, there is very little about where ly6-AMPAR interaction is expected. If the experiments are limited for whatever reason, it does not make much sense to look for possible interactions of lypd2 with Q or R variants of GluA2. It would make more sense to test other AMPAR subunits (A1, A3, A4, including heteromers) or complexes of AMPARs with auxiliary subunits. That would make the study more informative with the same level of experimentation.

The reason for this is predominantly extracellular location of ly6 proteins: they are either secreted or GPI-anchored to the outer leaflet of the plasma membrane whereas the Q/R editing site is in the pore of AMPARs, i.e. in the transmembrane region. It is not clear how the two could interact. Of course, the only way to rule this out is by experiments, however, if experiments are limited as it seems to be the case here, then it would make more sense to focus on variation in extracellular parts of AMPARs (provided by different AMPAR core subunits and complexes of AMPARs with auxiliary subunits) as these are more likely sites of interaction with lypd2 (ly6-related alpha-bungarotoxin also interacts with extracellular domains of acetylcholine receptors).

The authors even do pull down Stargazin and AMPAR complexes, but for some reason do not test these for the possible presence of lypd2.

The authors correctly note rich repertoire of AMPAR auxiliary subunits, however:

Line 76: please note that Stargazin is a TARP.

Lines 77-80 – it is not clear to me at all how AMPAR auxiliary proteins share cellular location and molecular architecture with ly6 proteins. Listed AMPAR auxiliary proteins are all transmembrane proteins whereas ly6 proteins are secreted or GPI-anchored to the outer leaflet of the plasma membrane. The authors state that AMPAR auxiliary subunits are heavily stabilized by disulphide bonds which is not the case. They provide a 2010 reference to support this statement, however, no structures of auxiliary subunits were available then. Meanwhile, structures of TARPs y2 (Stargazin), y8 and y5 came out as well as of CNIHs 2 and 3 which do not show extensive disulphide linking. Please, also note that AMPARs in the brain are almost always associated with auxiliary subunits (see e.g. Yu et al, Nature, 2021; https://doi.org/10.1038/s41586-021-03540-0), which means that if lypd2 was interacting with native AMPARs in hippocampus, it would have to do so with AMPARs complexed with auxiliary subunits (y8 and CNIH2).

RT-PCR is used to show expression of lypd2 in hippocampus and although AMPARs are also expressed in hippocampus, this does not strictly mean co-localization. This should be at least acknowledged in the discussion.

I don’t think the ratio of Stargazin – A2 DNA used in transfections is mentioned anywhere.

Maybe I missed it, but it is not mentioned anywhere how many times Western Blots or imaging experiments were replicated.

Figures:

The figures are quite low in resolution.

A panel or a figure to introduce the topic would be very helpful, e.g. AMPAR structure (including Q/R editing site) and a model of ly6 protein in the same membrane (UniProt entry for lypd2 provides alphafold structure which might be useful).

Raw images of gels relating to Fig. 1A are quite over-exposed. The amount of loaded ladder should be reduced to make it readable. Also, please explain what “X” stands for here.

Figure 1B – actin bands are very faint. Again, there seem to be some bands in a well marked “X”.

Figure 1C – why is over-expressed lypd2 visible as puncta? Is it aggregating?

Discussion is very limited and should be extendend (e.g. by acknowledging that the performed experiments do not entirely exclude interaction between lydp2 and AMPARs as other core subunits have not been tested nor complexes of AMPARs with auxiliary subunits).

Some typos:

Abstract: “alpha” shows up as “a”

Line 67 – delete “of”

Line 100 – was instead of were?

Line 120 – degree sign missing after 37

Line 122 – 2 mM

6. PLOS authors have the option to publish the peer review history of their article (what does this mean?). If published, this will include your full peer review and any attached files.

Reviewer #1: No

Reviewer #2: No

---

## [Author Response · Author response to Decision Letter 0]

24 Aug 2022

We thank the reviewers for their feedback. We have addressed the comments and requests for revisions from the expert reviewers, including an additional figure and text to clarify points of confusion (see below). In doing so, we believe we have significantly improved and strengthened our manuscript. In the following, we provide an itemized list of changes we have made in addressing the provided comments. 

Response to Reviewers

Reviewer 1:

“The task formulated by the authors - to detect novel receptors with which may interact the proteins of the Ly6/uPAR family - is well justified. They correctly started by emphasizing the well-known common three-finger folding of these proteins and of the three-finger alpha-neurotoxins from the snake venoms. However, they did clearly mention that there are two classes of the Ly6/uPAR proteins - secreted like SLURP-1 and membrane-bound by the GPI anchor like Lynx1. They gave the reasons why they have chosen lypd2 (because there are data on its colocalization with the AMPA receptors) but did not mention that lypd 2 is membrane-bound.”

We thank the reviewer for this careful attention to detail and have noted on line 126 that lypd2 is predicted to be a GPI-anchored protein. 

“The authors made several mistakes in the introduction concerning snake venom neurotoxins: line 49: major binding site of the snake venom alpha-neurotoxins is the orthosteric binding site (where agonists are bound), but not the allosteric sites (where mainly other allosteric positive or negative modulators are attached),”

We thank the reviewer for noting this error and have clarified on line 55 that alpha-neurotoxins primarily exert their effects through the orthosteric site, not the allosteric site.

“line 52: give references. line 53: alpha -bungarotoxin does not inhibit alpha4beta2 receptors, and instead of alpha7beta1 should be written "alpha7beta2.”

We have included additional references on line 58 on the earliest work studying lynx1 modulation of nicotinic acetylcholine receptors (references 19 and 20). We have corrected and clarified the effects of bungarotoxin on inhibition of homopentameric alpha7 receptors while lynx1 and lynx2 have modulatory effects on alpha7 and alpha4beta2 receptors on lines 59-61, and included an additional reference (reference 22).

“line 20 and through the whole text authors use the term "ly6 proteins" , although most often in the literature are used Ly6 or Ly6/uPAR proteins.”

We have gone through the manuscript and replaced “ly6 proteins” or “ly6” with “Ly6 proteins” and “Ly6.”

Reviewer 2:

“Overall, there is very little about where ly6-AMPAR interaction is expected. If the experiments are limited for whatever reason, it does not make much sense to look for possible interactions of lypd2 with Q or R variants of GluA2. If the experiments are limited for whatever reason, it does not make much sense to look for possible interactions of lypd2 with Q or R variants of GluA2. It would make more sense to test other AMPAR subunits (A1, A3, A4, including heteromers) or complexes of AMPARs with auxiliary subunits. That would make the study more informative with the same level of experimentation. The reason for this is predominantly extracellular location of ly6 proteins: they are either secreted or GPI-anchored to the outer leaflet of the plasma membrane whereas the Q/R editing site is in the pore of AMPARs, i.e. in the transmembrane region. It is not clear how the two could interact. Of course, the only way to rule this out is by experiments, however, if experiments are limited as it seems to be the case here, then it would make more sense to focus on variation in extracellular parts of AMPARs (provided by different AMPAR core subunits and complexes of AMPARs with auxiliary subunits) as these are more likely sites of interaction with lypd2 (ly6-related alpha-bungarotoxin also interacts with extracellular domains of acetylcholine receptors).”

We thank the reviewer for this feedback. We agree that it does not make sense to look for interactions of lypd2 with the amino acids at position 607, representing the Q/R editing site on GluR2, and it was not our original intention to identify or propose direct interactions between lypd2 and the Q/R amino acid site. To help clarify this confusion, we have:

1) Included Figure 1 (lines 117-123) at the reviewer’s suggestion below to highlight where we expect Ly6-GluR2 receptor interactions to occur at the structural level. In this model, based on the scale of each molecule relative to each other, it is likely that a Ly6 protein could make contact with the ligand binding domain and/or the linker between the ligand binding domain and the transmembrane domain. 

2) Striven to clarify our logic with regards to why we chose to focus on the Q and R variants of the GlRA2 homomer in our study in lines 131-166. Our interest in these isoforms was on the basis of literature indicating isoform-specific differences in interactions and regulations with small regulatory proteins such as the type-II TARPs �-5 and �-7. 

3) We have also expanded the discussion (lines 322-328) to acknowledge that there are other AMPAR variations in terms of subunit composition and editing events that we did not explore that could also be possibilities for interaction.

“The authors even do pull down Stargazin and AMPAR complexes, but for some reason do not test these for the possible presence of lypd2.”

Given that AMPAR receptors have been shown to participate in dimeric complexes (detected as an interaction between the receptor and a single auxiliary protein of interest using a transfected cell culture system absent other auxiliary proteins), our goal in this study was to delineate the possibility of an interaction between Ly6 proteins and AMPARs using a similar approach. Consequently, the purpose of the stargazin pulldown was to serve as a positive control for our pulldown conditions, not for the purposes of detecting a trimeric complex. We thank the reviewer for this feedback, however, and we highlight in our expanded discussion in lines 327-336 that it is certainly possible that there could be trimeric complexes between AMPAR, Ly6 proteins, and these auxiliary proteins and indicate that this could be a source of future exploration.

“The authors correctly note rich repertoire of AMPAR auxiliary subunits, however: Line 76: please note that Stargazin is a TARP.”

We thank the reviewer for this careful attention to detail and have noted on line 97 that stargazin is a TARP.

“Lines 77-80 – it is not clear to me at all how AMPAR auxiliary proteins share cellular location and molecular architecture with ly6 proteins. Listed AMPAR auxiliary proteins are all transmembrane proteins whereas ly6 proteins are secreted or GPI-anchored to the outer leaflet of the plasma membrane. The authors state that AMPAR auxiliary subunits are heavily stabilized by disulphide bonds which is not the case. They provide a 2010 reference to support this statement, however, no structures of auxiliary subunits were available then. Meanwhile, structures of TARPs y2 (Stargazin), y8 and y5 came out as well as of CNIHs 2 and 3 which do not show extensive disulphide linking.”

We thank the reviewer for this feedback, and have updated both our text in the manuscript and our references accordingly. The new structural information available on AMPAR-auxiliary protein structures is very exciting, and indeed shifts our perspective on the similarities between these auxiliary proteins and Ly6s—as we have modified in the text, it is evident that the similarities are less in molecular structure and more in the size and placement of these small regulatory proteins. Both Ly6s, TARPS, and CNIHs are located in close proximity to the plasma membrane, which limits their contact points to largely the ligand binding domain and/or the linker region between the ligand binding domain and the transmembrane domain of the AMPAR. We have clarified our language in the introduction to reflect this in lines 99-111 and 161-163.

“Please, also note that AMPARs in the brain are almost always associated with auxiliary subunits (see e.g. Yu et al, Nature, 2021; https://doi.org/10.1038/s41586-021-03540-0), which means that if lypd2 was interacting with native AMPARs in hippocampus, it would have to do so with AMPARs complexed with auxiliary subunits (y8 and CNIH2). RT-PCR is used to show expression of lypd2 in hippocampus and although AMPARs are also expressed in hippocampus, this does not strictly mean co-localization. This should be at least acknowledged in the discussion.”

We thank the reviewer for this thoughtful feedback. We have expanded the discussion in lines 322-336 to acknowledge the questions left unexplored in our current study and to consider these possibilities as avenues for future exploration.

“I don’t think the ratio of Stargazin – A2 DNA used in transfections is mentioned anywhere.”

We have updated the figure legend on line 303-304 for Figure 3 with the ratio of transfected DNAs.  

“Maybe I missed it, but it is not mentioned anywhere how many times Western Blots or imaging experiments were replicated.”

All figures for Westerns and imaging experiments are representative images from at least three replicates; we have noted this in lines 225-226 and 247-248 in the methods. 

“The figures are quite low in resolution.”

The figures were created using digital image files saved from our GE ImageQuant Western Blot Imager. We note the reviewer’s feedback, but unfortunately are only able to work with the image resolution from the saved images. We will examine the settings on our instrument to see if there is the possibility of improving the image resolution for future work. 

“A panel or a figure to introduce the topic would be very helpful, e.g. AMPAR structure (including Q/R editing site) and a model of ly6 protein in the same membrane (UniProt entry for lypd2 provides alphafold structure which might be useful).”

We thank the reviewer for their feedback and have created Figure 1 to show a juxtaposition of structural data for GluR2 and lynx1 as well as the Alphafold model of lypd2. Since the structure of lynx1 has been experimentally determined, we chose to use lynx1 in panel A to maintain accuracy in scale and structural detail; however, as shown in panel B it is likely that lypd2 shares significant structural homology to lynx1. We have introduced and explained this model in the text as well, in lines 117-123 and 155-163.  

“Raw images of gels relating to Fig. 1A are quite over-exposed. The amount of loaded ladder should be reduced to make it readable. Also, please explain what “X” stands for here.”

 “X” refers lanes containing other samples not relevant to this publication and cropped out of the final figures in the manuscript; we have appended the labeling associated with our raw image files to reflect this annotation. We have also included an additional exposure of the gel for the whole brain sample in which the ladder was not overexposed to demonstrate the correct size of the expected PCR products.

 “Figure 1B – actin bands are very faint. Again, there seem to be some bands in a well marked “X”.”

We note the reviewer’s critique here, and acknowledge that the actin bands are very faint. However, we do not have the ability to reexpose that specific blot as the physical blot itself was discarded after those initial exposures. The lanes marked X have been previously described in the legend for Fig 1A, and apply to all subsequent figures.

 “Figure 1C – why is over-expressed lypd2 visible as puncta? Is it aggregating?”

GPI anchored Ly6 proteins have been previously observed to form puncta, possibly due to localization to microdomains in the plasma membrane. It is thought that these puncta represent functional organization of the Ly6 proteins at the plasma membrane (in contrast to non-functional aggregates), although additional details remain unknown. We have included additional references in the text (line 261). 

“Discussion is very limited and should be extendend (e.g. by acknowledging that the performed experiments do not entirely exclude interaction between lydp2 and AMPARs as other core subunits have not been tested nor complexes of AMPARs with auxiliary subunits).”

We thank the reviewer for this feedback and have expanded the discussion in lines 323-337 to highlight the these potential avenues of future exploration with regards to other AMPARs containing other core subunits as well as the possibility of trimeric complexes between AMPARs, auxiliary subunits, and lypd2.

“Some typos: Abstract: “alpha” shows up as “a” Line 67 – delete “of” Line 100 – was instead of were? Line 120 – degree sign missing after 37 Line 122 – 2 mM”

We thank the reviewer for such a close reading of the text, and these typos have been addressed in the revised manuscript text.

---

## [Decision Letter · Decision Letter 1]

12 Sep 2022

PONE-D-22-12127R1GluR2Q and GluR2R AMPA Subunits are not Targets of lypd2 InteractionPLOS ONE

Dear Dr. Sung,

Thank you for submitting your manuscript to PLOS ONE. After careful consideration, we feel that it has merit but does not fully meet PLOS ONE’s publication criteria as it currently stands. Therefore, we invite you to submit a revised version of the manuscript that addresses the points raised during the review process. As I wrote under Comments to the Author, Reviewer 2 suggests an extensive set of experiments that I do not think that you need to execute in the present manuscript. The reviewer does, however, raise several, more minor, issues that you are requested to address as fully as possible in your revised version. Please submit your revised manuscript by Oct 27 2022 11:59PM. If you will need more time than this to complete your revisions, please reply to this message or contact the journal office at plosone@plos.org. Please include the following items when submitting your revised manuscript:A rebuttal letter that responds to each point raised by the academic editor and reviewer(s). You should upload this letter as a separate file labeled 'Response to Reviewers'.A marked-up copy of your manuscript that highlights changes made to the original version. You should upload this as a separate file labeled 'Revised Manuscript with Track Changes'.An unmarked version of your revised paper without tracked changes. You should upload this as a separate file labeled 'Manuscript'.If applicable, we recommend that you deposit your laboratory protocols in protocols.io to enhance the reproducibility of your results. Protocols.io assigns your protocol its own identifier (DOI) so that it can be cited independently in the future. For instructions see: https://journals.plos.org/plosone/s/submission-guidelines#loc-laboratory-protocols. Additionally, PLOS ONE offers an option for publishing peer-reviewed Lab Protocol articles, which describe protocols hosted on protocols.io. Read more information on sharing protocols at https://plos.org/protocols?utm_medium=editorial-email&utm_source=authorletters&utm_campaign=protocols.

We look forward to receiving your revised manuscript.

Kind regards,

Israel Silman

Academic Editor

PLOS ONE

Journal Requirements:

Additional Editor Comments:

As you will see, Reviewer 2 has provided an extensive review which suggests that you perform many additional experiments. Although the experiments suggested are constructive, and worth executing, I do not think that you need to address them in the present manuscript. The reviewer does, however, make several more minor criticisms that you should try to address as fully as possible in your revised manuscript.

Reviewers' comments:

Reviewer's Responses to Questions

**Comments to the Author**

1. If the authors have adequately addressed your comments raised in a previous round of review and you feel that this manuscript is now acceptable for publication, you may indicate that here to bypass the “Comments to the Author” section, enter your conflict of interest statement in the “Confidential to Editor” section, and submit your "Accept" recommendation.

Reviewer #1: All comments have been addressed

Reviewer #2: (No Response)

2. Is the manuscript technically sound, and do the data support the conclusions?

Reviewer #1: Yes

Reviewer #2: Partly

3. Has the statistical analysis been performed appropriately and rigorously? 

Reviewer #1: I Don't Know

Reviewer #2: No

4. Have the authors made all data underlying the findings in their manuscript fully available?

Reviewer #1: Yes

Reviewer #2: Yes

5. Is the manuscript presented in an intelligible fashion and written in standard English?

Reviewer #1: Yes

Reviewer #2: Yes

6. Review Comments to the Author

Reviewer #1: I am glad that my critical comments were taken into account and the required changes were introduced into the revised version.

Reviewer #2: I thank the authors for going through all my points and in particular for including a new figure (Fig. 1) as I know this can take time.

I do think the study addresses an interesting question and provides a result (lypd2 does not interact with GluA2 homomers), but my main concerns still remain and that is the logic behind the experiments and comparisons the authors draw between Ly6 and AMPAR auxiliary subunits.

Yes, they are both (Ly6 and auxiliary proteins) associated with membrane, but all AMPAR auxiliary proteins are transmembrane proteins and interact fairly tightly with AMPAR transmembrane regions. Hence, their interactions with AMPARs can be expected to be sensitive to Q/R editing site in the pore. Ly6 proteins are anchored to the membrane, but are not transmembrane proteins. Their interaction points are most likely to be with the extracellular domains of AMPARs as shown in Fig. 1.

So why not test then other AMPARs subunits and/or variants which differ in the extracellular region and not just in the pore region?

Yes, Q/R editing site is very important for kinetics, trafficking, assembly etc. but majority of these come from the fact that this residue is in the pore and directly determines calcium permeability and pore properties.

Currently the logic is: protein A and protein B perhaps interact across this interface. To test this interaction, we will use mutation in A away from the interaction interface as it is known to be important for A.

Why not find examples of compounds/peptides that interact with extracellular domains of AMPARs and differ between Q and R variants (I am not aware of any, but I might be wrong)? Or perhaps there is a mutation in the pore region of AChRs that affects binding of alpha-bungarotoxin? If so, this would introduce well the rest of the paper.

"Given that AMPAR receptors have been shown to participate in dimeric complexes (detected as an interaction between the receptor and a single auxiliary protein of interest using a transfected cell culture system absent other auxiliary proteins), our goal in this study was to delineate the possibility of an interaction between Ly6 proteins and AMPARs using a similar approach. Consequently, the purpose of the stargazin pulldown was to serve as a positive control for our pulldown conditions, not for the purposes of detecting a trimeric complex."

AMPARs have also been expressed as a trimeric complex in a heterologous system (A1 + y8 + CNIH-2 in HEK cells in Kato et al., Neuron, 2010; https://doi.org/10.1016%2Fj.neuron.2010.11.026; PMID: 21172611), so still not sure it makes sense to use it only as a pulldown control and not in an experiment which would substantially strengthen the paper (in my opinion).

Minor comments:

1) Thank you for making it clear that n is 3 or more for all experiments, but it would be good to specify the exact number of replicates (n = x) for each experiment. Some quantification of at least some of the experiments would also strengthen the paper.

2) Figure 1 – nice figure! Orienting the two peptides in the same way would help comparison and it would also make it clearer how similar the fold is.

3) Line 104-105: "AMPA receptors are tetrameric and can exist as heteromeric combinations of GluR1 with either GluR2 or GluR3 or as homomers of either GluR1 or GluR2."

To me this reads like AMPARs can form A1/A3 heteromers. At least in vivo, much more dominant heterotetramer combinations are A1/A2/A1/A2, A3/A2/A3/A2 and A1/A2/A3/A2 (see e.g. Yu et al, Nature, 2021; https://doi.org/10.1038/s41586-021-03540-0).

4) "Several auxiliary proteins, including TARPs (such as stargazin), GRIP2/ABP, PICK1, and CNIHs and stargazin, have already been identified as playing key roles in AMPAR trafficking to the cell surface (38–40)."

GRIP2 and PICK1 do interact with AMPARs, but are not considered to be their auxiliary subunits in the way TARPs (and others) are (Schwenk et al., Neuron, 2012).

7. PLOS authors have the option to publish the peer review history of their article (what does this mean?). If published, this will include your full peer review and any attached files.

Reviewer #1: No

Reviewer #2: No

---

## [Author Response · Author response to Decision Letter 1]

25 Oct 2022

Reviewer #1: 

I am glad that my critical comments were taken into account and the required changes were introduced into the revised version.

Reviewer #2: 

I thank the authors for going through all my points and in particular for including a new figure (Fig. 1) as I know this can take time.

We thank both reviewers for their attention to our manuscript, and we feel the manuscript has been significantly improved on the basis of their feedback. 

Reviewer #2

“Yes, they are both (Ly6 and auxiliary proteins) associated with membrane, but all AMPAR auxiliary proteins are transmembrane proteins and interact fairly tightly with AMPAR transmembrane regions. Hence, their interactions with AMPARs can be expected to be sensitive to Q/R editing site in the pore. Ly6 proteins are anchored to the membrane, but are not transmembrane proteins. Their interaction points are most likely to be with the extracellular domains of AMPARs as shown in Fig. 1.”

We respectfully disagree with the expectation that the interaction between the transmembrane domains of AMPARs and auxiliary proteins are the molecular basis for their regulation of AMPAR function. The interaction between transmembrane domains may be important for complex formation and stability, but prior work mapping the functional interface between TARPs and AMPARs have shown that 1) the extracellular loops of TARPs mediate their functional effects on AMPARs and 2) the regions important for mediating these effects on AMPARs include the linker region between the N-terminal domain (NTD) and ligand binding domain (LBD) for �-2 and the linker region between the LBD and the transmembrane domain (TMD) for �-8 1,2,3. Ly6 proteins, despite not having a transmembrane domain, are positioned well to interact with any of these linker regions to potentially mediate functional impacts in a similar mechanism as the extracellular loops of TARPs.

1Ben-Yaacov A, Gillor M, Haham T, Parsai A, Qneibi M, Stern-Bach Y. Molecular Mechanism of AMPA Receptor Modulation by TARP/Stargazin. Neuron. 2017 Mar;93(5):1126-1137.e4.

2Cais O, Herguedas B, Krol K, Cull-Candy SG, Farrant M, Greger IH. Mapping the Interaction Sites between AMPA Receptors and TARPs Reveals a Role for the Receptor N-Terminal Domain in Channel Gating. Cell Rep. 2014 Oct;9(2):728–40.

3Herguedas B, Watson JF, Ho H, Cais O, García-Nafría J, Greger IH. Architecture of the heteromeric GluA1/2 AMPA receptor in complex with the auxiliary subunit TARP γ8. Science. 2019 Apr 26;364(6438).

We have added additional language in the introduction (lines 81-87 and 124-128) to 1) clarify that auxiliary protein mediated regulation of AMPARs are likely to be mediated through contacts between the TMD-LBD or LBD-NBD linkers, not direct contact with the Q/R site and 2) include the additional citations above. 

“So why not test then other AMPARs subunits and/or variants which differ in the extracellular region and not just in the pore region?”

Our experiments do not rule out the possibility of other domains on the AMPAR as playing a role in the interaction. We have included language in the discussion (lines 292-293) to clarify this; we describe the logic of our reasoning in a response to a separate reviewer comment further below. 

“Yes, Q/R editing site is very important for kinetics, trafficking, assembly etc. but majority of these come from the fact that this residue is in the pore and directly determines calcium permeability and pore properties.”

The biochemical mechanism behind the role of the Q/R editing site in determining calcium permeability and pore properties is quite clear and has been previously established; however, this only encapsulates direct effects post-assembly once the full pore has been established within a correctly folded and assembled AMPAR tetramer. It has also been shown that the editing at this site plays a critical role in the steps of AMPAR maturation prior to complete assembly. Greger et al4 shows changes in export kinetics from the ER between Q and R forms, indicating that the change in amino acid at this position has potential impacts on AMPAR function at a stage in its biogenesis where pore formation may not have occurred (i.e during protein folding and/or assembly). Moreover, Cais et al2 observed Q/R isoform-specific differences in association between �-2 and AMPARs lacking an NTD (decreased co-immunoprecipitation with GluR2R vs GluR2Q), suggesting that the effects of the Q/R editing site extend beyond direct impacts to the pore.

4Greger IH, Khatri L, Ziff EB. RNA Editing at Arg607 Controls AMPA Receptor Exit from the Endoplasmic Reticulum. Neuron. 2002 May;34(5):759–72.

Consequently, the role of the Q/R site in AMPAR function extends beyond its direct function in the pore to include indirect effects on AMPAR biogenesis as well. We have added language to 1) distinguish the effects of the Q/R editing on immature vs mature AMPAR (lines 117-120) and 2) highlight that any mechanism of isoform specific regulation is unlikely to occur through direct contact with the Q/R site itself (line 154).

“Currently the logic is: protein A and protein B perhaps interact across this interface. To test this interaction, we will use mutation in A away from the interaction interface as it is known to be important for A.”

We would like to clarify that our logic is the following (using similar language as the reviewer):

1) Protein A (AMPAR) and protein B (Ly6) could interact at the cell surface. As shown in Fig 1, the interaction interface is likely to be extracellular due to predicted anchoring of Ly6 proteins to the extracellular face of the membrane.

2) Prior work on other family members for protein B suggest roles in regulation of ionotropic receptor function. Prior work on Protein A (an ionotropic receptor) demonstrates multiple examples of regulation via extracellular interactions with small regulatory proteins. 

3) Specifically, Protein A has different isoforms and combinations that each have unique biochemical properties and biological functions. One type of regulation is the effects of small regulatory proteins on properties of homomeric Protein AQ and Protein AR.

4) As there has been no prior work investigating the possibility of interaction and/or regulation between Protein A and Protein B, we choose to begin our work by examining the simplest molecular version of Protein A that still has relevance for function: homomeric Protein AQ and Protein AR.

Our hypothesis: If protein B is involved in regulation of function of protein AQ and protein AR, then protein B may interact differently with protein AQ and protein AR.

As noted in our previous revision, we found Reviewer #2’s comments very helpful in clarifying our hypothesis and we included substantial changes to our text to highlight that our hypothesis does not rule out the possibility of protein B interacting or regulating other forms of protein A. Given that there was no information available on regulation of protein A by protein B, we began with this hypothesis with these two isoforms as a simple starting point in our investigations (working with a homomeric receptor with only one amino acid change) that would still contribute knowledge to the field (working with isoforms that have been previously studied and have significant functional implications). Consequently, our choice to study the Q/R isoforms was not to “use the mutation to test the interaction interface,” but rather as a simpler system for evaluating potential effects on AMPAR function. While we understand there can be a variety of approaches towards investigating a completely uncharacterized system (here, Ly6 protein interaction with AMPAR), we hope this clarifies our approach to the reviewer. We have included additional language in lines 108-118 to clarify this in the text.

“Why not find examples of compounds/peptides that interact with extracellular domains of AMPARs and differ between Q and R variants (I am not aware of any, but I might be wrong)?” 

To our knowledge, we are also unable to identify the reagents the reviewer is suggesting, and so would be unable to do this experiment. 

“Or perhaps there is a mutation in the pore region of AChRs that affects binding of alpha-bungarotoxin? If so, this would introduce well the rest of the paper.”

To our knowledge there is not a mutation in the pore region of nAChRs that affects binding of bungarotoxin. We did find a work studying the influence of a pore mutation T288N on GABAA receptor trafficking5; however, this was absent data related to any potential impacts on auxiliary and/or small regulatory protein function.

5Hernandez, C.C., Zhang, Y., Hu, N. et al. GABA A Receptor Coupling Junction and Pore GABRB3 Mutations are Linked to Early-Onset Epileptic Encephalopathy. Sci Rep 7, 15903 (2017). https://doi.org/10.1038/s41598-017-16010-3.

"AMPARs have also been expressed as a trimeric complex in a heterologous system (A1 + y8 + CNIH-2 in HEK cells in Kato et al., Neuron, 2010; https://doi.org/10.1016%2Fj.neuron.2010.11.026; PMID: 21172611), so still not sure it makes sense to use it only as a pulldown control and not in an experiment which would substantially strengthen the paper (in my opinion).”

We acknowledge and appreciate the reviewer’s opinion on this experiment, and have included the new citation in line 294. As we noted in our previously revised text, our experiments do not exclude the possibility of a trimeric complex and that is a source for future work. 

“1) Thank you for making it clear that n is 3 or more for all experiments, but it would be good to specify the exact number of replicates (n = x) for each experiment. Some quantification of at least some of the experiments would also strengthen the paper.”

We have updated the specific n for each experiment in the figure legends. With regards to quantification—we are unsure as to which experiments the reviewer is referring to. For example, while we recognize the value in quantifying Western Blot intensities for certain experiments (for example, quantifying differences in surface expression for surface biotinylation or labeling experiments), given that our results did not show an interaction we are unsure what the benefit of quantifying those bands (or in our case, the lack of bands) would be.

“2) Figure 1 – nice figure! Orienting the two peptides in the same way would help comparison and it would also make it clearer how similar the fold is.”

Thank you. The convention for orienting structures of Ly6 proteins in the literature is generally with the disulfide bonds at the “top” of the image and the three loops facing “down;” we have maintained this orientation in panel B to maintain consistency with depictions in the field. To help the reader orient, we intentionally kept the color and representation of the disulfide bonds (gold sticks) consistent between Lynx1 in panel A and the model of lypd2 in panel B to facilitate comparison between the two proteins. 

“3) Line 104-105: "AMPA receptors are tetrameric and can exist as heteromeric combinations of GluR1 with either GluR2 or GluR3 or as homomers of either GluR1 or GluR2."

To me this reads like AMPARs can form A1/A3 heteromers. At least in vivo, much more dominant heterotetramer combinations are A1/A2/A1/A2, A3/A2/A3/A2 and A1/A2/A3/A2 (see e.g. Yu et al, Nature, 2021; https://doi.org/10.1038/s41586-021-03540-0).”

We thank the reviewer for their close reading of the text, and have adjusted the language in lines 104-105 to be clearer. 

“4) "Several auxiliary proteins, including TARPs (such as stargazin), GRIP2/ABP, PICK1, and CNIHs and stargazin, have already been identified as playing key roles in AMPAR trafficking to the cell surface (38–40)."

GRIP2 and PICK1 do interact with AMPARs, but are not considered to be their auxiliary subunits in the way TARPs (and others) are (Schwenk et al., Neuron, 2012).”

We have adjusted the language in lines 76-78 to reflect this and included two additional citations (both the reviewer’s suggestion and the one below).

Bissen D, Foss F, Acker-Palmer A. AMPA receptors and their minions: auxiliary proteins in AMPA receptor trafficking. Cell Mol Life Sci. 2019 Jun;76(11):2133–69.

---

## [Decision Letter · Decision Letter 2]

14 Nov 2022

GluR2Q and GluR2R AMPA Subunits are not Targets of lypd2 Interaction

PONE-D-22-12127R2

Dear Dr. Sung,

We’re pleased to inform you that your manuscript has been judged scientifically suitable for publication and will be formally accepted for publication once it meets all outstanding technical requirements.

Kind regards,

Israel Silman

Academic Editor

PLOS ONE

Additional Editor Comments (optional):

Reviewers' comments:

Reviewer's Responses to Questions

**Comments to the Author**

1. If the authors have adequately addressed your comments raised in a previous round of review and you feel that this manuscript is now acceptable for publication, you may indicate that here to bypass the “Comments to the Author” section, enter your conflict of interest statement in the “Confidential to Editor” section, and submit your "Accept" recommendation.

Reviewer #2: All comments have been addressed

2. Is the manuscript technically sound, and do the data support the conclusions?

Reviewer #2: Yes

3. Has the statistical analysis been performed appropriately and rigorously? 

Reviewer #2: N/A

4. Have the authors made all data underlying the findings in their manuscript fully available?

Reviewer #2: Yes

5. Is the manuscript presented in an intelligible fashion and written in standard English?

Reviewer #2: Yes

6. Review Comments to the Author

Reviewer #2: I thank the authors for addressing all of my comments. I think the manuscript reads much better now.

7. PLOS authors have the option to publish the peer review history of their article (what does this mean?). If published, this will include your full peer review and any attached files.

Reviewer #2: No

---

## [Editor Report · Acceptance letter]

16 Nov 2022

PONE-D-22-12127R2 

GluR2Q and GluR2R AMPA Subunits are not Targets of lypd2 Interaction 

Dear Dr. Sung:

I'm pleased to inform you that your manuscript has been deemed suitable for publication in PLOS ONE. Congratulations! Your manuscript is now with our production department. 

Kind regards, 

on behalf of

Prof. Israel Silman 

Academic Editor

PLOS ONE